# The Identification of Elderly People with High Fall Risk Using Machine Learning Algorithms

**DOI:** 10.3390/healthcare11010047

**Published:** 2022-12-23

**Authors:** Ziyang Lyu, Li Wang, Xing Gao, Yingnan Ma

**Affiliations:** 1Institute of Smart Ageing, Beijing Academy of Science and Technology, Beijing 100035, China; 2School of Biological Science and Medical Engineering, Beihang University, Beijing 100191, China

**Keywords:** elderly, fall risk, multifractal algorithm, machine learning

## Abstract

Falling is an important public health issue, and predicting the fall risk can reduce the incidence of injury events in the elderly. However, most of the existing studies may have additional human and financial costs for community workers and doctors. Therefore, it is socially important to identify elderly people who are at high fall risk through a reasonable and cost-effective method. We evaluated the potential of multifractal, machine learning algorithms to identify the elderly at high fall risk. We developed a 42-point calibration model of the human body and recorded the three-dimensional coordinate datasets. The stability of the motion trajectory is calculated by the multifractal algorithm and used as an input dimension to compare the performance of the six classifiers. The results showed that the instability of the faller group was significantly greater than that of the no-faller group in the male and female cohorts (*p* < 0.005), and the Gradient Boosting Decision Tree classifier showed the best performance. The findings could help elderly people at high fall risk to identify individualized risk factors and facilitate tailored fall interventions.

## 1. Introduction

Falling is the major cause of trauma, disability, and fatal events in the elderly and places a high economic strain on individuals, families, and social public health systems [1,2]. Studies have shown that individuals can significantly reduce fall risk by participating in fall prevention programs [3,4]. Therefore, it is important to conduct extensive fall risk screening in the elderly. Fall prevention aims to predict fall risk in advance through some movement characteristics of the human body, and help medical personnel take timely intervention measures.

Timed Up and Go (TUG) is a tool that is widely used to screen the elderly for the high fall risk [5]. The elderly people walk on a 3-m footpath and then turn back to the starting point. When the total time exceeds a time threshold, it is considered a high fall risk [6]. TUG experiments use process time as a classification parameter, and these reference values are often derived from statistical values of a limited sample, and in a wide range of applications, the range of time thresholds is increasingly controversial.

For example, guidelines from the American Geriatrics Society (AGS) and the British Geriatrics Society (BGS) recommend a TUG threshold time of 13.5 s for the prediction of fall risk in elderly people in the community [7], but different recommendations have been made in some studies, on the one hand, there may be some error in the measurement process and, on the other hand, the confidence in the use of this threshold has shown a large variation [8,9,10]. Additionally, there are geographical and ethnographic differences in the threshold time of TUG [11] which increases the potential misclassification rate by using time as a classification criterion.

Therefore, researchers try to find more sensitive indicators that can be used for fall risk prediction, Multimodal gait analysis technology is introduced in the TUG process to make up for the shortage of using time threshold as the fall risk assessment standard. In recent years, it is also the main research direction to determine the sensitivity index through the posture data. It has been suggested that the area of sway per unit of time, anterior-posterior average velocity, and radial average velocity of the human center of pressure (CoP) are the best indicators of sensitivity [12,13]. Some studies have also used a combination of linear and non-linear features of motion trajectories as sensitivity indicators [14]. However, the accuracy of these metrics has yet to be extensively validated. In past studies, the gait signal is usually treated as a regular periodic signal, but it has been pointed out that human motion is a nonlinear process and the gait signal is quasi-periodic [15,16]. The scalar of nonlinear dynamical systems is usually described using fractal geometry [17]. Fractal geometry is a method used to analyze nonlinear curves and complex images, and in past studies, nonlinear signals such as EEG [18], EMG [19], and spectra [17] were analyzed, and multiple fractal algorithms decompose the original signal into multiple subsets of different fractal dimensions, which can better describe the irregular variations of the signal.

Also, based on the identification of sensitive indicators, the use of machine learning algorithms enables the analysis of additional features of the indicator dataset and gives decision results. Machine learning algorithms have been well applied in the field of fall risk of the elderly [20,21,22], and can make full use of historical data features to give prediction results. Some research [23] identified fallers by tree-based algorithms for a group of 141 elderly with a mean age of 73.2 ± 11.4 years, based on a baseline eye examination and standardized questionnaires including lifestyle, general health, social participation, and vision problems, and obtained an accuracy of 75.9%. Some research [24] recorded movement data from tri-axial accelerometers in the head, pelvis, and left and right calves for a group of 100 elderly people with a mean age of 75.5 ± 6.7 years, identified fallers by support vector machines, neural networks, and other classifiers, and obtained 84% accuracy.

Although the use of machine learning is relatively widespread in the field of fall prevention, these machine-learning models are often regarded as black-box algorithms. The models lack a certain degree of interpretability, leading to a lack of understanding of the underlying principles behind the decision-making mechanism, and the accuracy of the recognition needs to be further improved. Therefore, the focus of this study is to develop an interpretable machine learning algorithm and identify kinematics parameters related to high fall risk. The purpose of this study is: (1) to evaluate the stability of joints in the process of human movement, and to analyze the difference in motion stability between different groups (2) based on machine learning and interpretable feature dimensions, a recognition framework for automatic recognition of fall risk in the elderly is developed.

## 2. Materials and Methods

### 2.1. Study Design, Participants, and Ethics

All 46 subjects included in this study were from a community in Beijing. Over the past year, they have been included in a community focus list because of their age and fall problems.

Inclusion and exclusion criteria: over the past five years, the community has created a dynamic focus list; as of this study, there were 98 people on the list. Forty of them had musculoskeletal system problems such as hip fractures and muscle contusions due to falls; five were unable to participate in the experiment due to the effects of COVID-19, and seven were excluded due to the quality of the experimental data. The 46 subjects included in the study were representative of the level of exercise among the elderly in this community. All subjects were older than 60 years old. They had no self-reported neurological diseases that affected walking, no musculoskeletal diseases that seriously affected walking, and no need for walking aids. The study was approved by the Ethics Review Committee of the Institute of Smart Ageing, Beijing Academy of Science and Technology, and written consent was obtained from all subjects before participation.

The Wilcoxon rank sum test did not reveal any significant differences between their basic information (age, BMI). 

Falling was defined according to the codes MB46.3 (A sudden spontaneous fall while standing and recovery within seconds or minutes) and MB47.C (Tendency to fall because of old age or other unclear health problems) from the International Classification of Diseases-11 (ICD-11).

The fall risk referred to in this study is based on fall history [25], which is directly related to the recurrence of falling. Therefore, we divided subjects with different fall risks into two groups: faller, and nofaller. Baseline characteristics for all subjects are shown in Table 1.

Subjects were tested with TUG, a motion capture system as motion analysis, 15 cameras in the experimental environment, sampling frequency 60 Hz, and we built a 42-Markers model to record motion in the sagittal, coronal, and cross-sectional planes, respectively, during the motion trajectory. The calibration model is shown in Figure 1.

### 2.2. Multifractal Algorithm

Define the nonlinear time series of the motion trajectory curve as ${\left\{x_{i}\right\}}_{i=1}^{n}$ and then transform it into a new series as ${\left\{x(k)\right\}}_{k=1}^{n}$.
(1)$$x(k)=\sum_{i=1}^{k}\left(x_{n}-\overset{-}{x}\right)k=1,2,\ldots n$$

Select a positive integer S and set it as the partition distance, the new series would then be divided into L=N/S segments (rounded down to the nearest integer). Define the set of segments as *v* = (1,2,…,*L*), fit the Vth curve with *f*(*v*) and figure out local detrending covariance [26]:(2)$$F^{2}(s,v)=\frac{1}{s}\sum_{j=1}^{s}{\left\{x[(v-1)s+j]-f_{v}(j)\right\}}^{2}$$

Figure out the *q*th-order detrending covariance with a given non-zero real number *q*:(3)$$\begin{array}{l} F_{q}(s)={\left\{\frac{1}{N_{S}}\sum_{v=1}^{N_{S}}{\left[F^{2}(s,v)\right]}^{\frac{q}{2}}\right\}}^{\frac{1}{q}}\\ if(q=0)\\ F_{0}(s)=e^{\frac{\left[\sum_{v=1}^{s}InF_{v}(s)\right]}{2s}}\\ F_{q}(s)\sim s^{h(q)} \end{array}$$

*h*(*q*) in the equation refers to generalized Hurst exponents. The Hurst exponents of the 46 groups of subjects have been calculated, which are not fixed but vary dynamically over time. The motion trajectory curve as a result displays multifractal features. The multifractal indicator τ(q) is defined as [27]:(4)$$\begin{array}{l} \tau (q)=qh(q)-1\\ \alpha =\frac{d(\tau (q))}{dq}\\ f(\alpha )=q\alpha -\tau (q) \end{array}$$

In the equation, α refers to the singularity exponent and *f*(*α*) refers to the multifractal spectrum.

Generalized fractal parameters have been added and defined as follows:

*α*_min_ refers to the minimum point of the fractal spectrum; *α*_max_ refers to the maximum point of the fractal spectrum; *α*_0_ refers to the singularity exponent corresponding to the maximum point of the fractal spectrum; Δ*α* refers to the width of the fractal spectrum in positive correlation with the fractal features of the figure and is defined as Δ*α* = *α*_max_ − *α*_min_; Δ*f* refers to fractal dimension difference between the maximum probability subset and the minimum probability subset and is defined as Δ*f = f*(*α*_min_) − *f*(*α*_max_), Δ*f >* 0, showing higher probability of sharp curve fluctuation than that of slight curve fluctuation; the stability of the cumulative distribution function is defined as:(5)$$D(\alpha )=|\alpha_{\max}-\alpha_{0}|-\left|\alpha_{\min}-\left.\alpha_{0}\right|\right.$$

The lower *D*(*α*) becomes, the more stable the cumulative distribution function would be.

### 2.3. Statistical Analyses

Significance analysis was performed using SPSS version 19.0 (IBM Corp, Armonk, NY, USA). Two types of comparisons were designed; one was to calculate the stability of all joint spots, define the stability of posture as the average of the stability of each joint spot, and calculate the relationship between gender differences [28] and fall risk. The other is a direct comparison of the stability of all joints in relation to the fall risk [29,30,31]. Both comparisons were performed using independent samples *t*-tests.

### 2.4. Machine Learning Approach

This study used L1/2 sparse iteration, Support vector machine (SVM), Gradient Boosting Decision Tree (GBDT), Random Forest (RF), Deep Neural Network (DNN), and Recurrent Neural Network (RNN) to predict the fall risk.

The design of RNN and CNN classifiers was performed in the software framework of TensorFlow version 2.10 (Google, Mountain View, CA, USA), and the design of other classifier methods was performed in the software framework of Matlab version r2022b (Mathworks, Natick, MA, USA). The parameter settings of all classifiers are shown in Table 2.

## 3. Results

### 3.1. Multifractal Singular Spectrum

The calculated results are shown in Figure 2, and the fractal spectrum shows an inverted bell shape, which proves that the motion trajectory has fractal characteristics.

### 3.2. Data Characteristics

The stability of 126 sets of motion trajectories of 42 markers in sagittal, coronal, and transverse planes was calculated, and the frequencies of the results distributed in different intervals were counted, and the distribution of the results is shown in Figure 3.

The data are distributed in the interval range of −2.5 to 11.5, where −0.5 to 0.5 is the main distribution interval, and the distribution of each marker is different, and these data contain the subtle variability of individuals in the process of movement.

### 3.3. Differences between Groups

As shown in Figure 4, the mean value of *D*(*α*) is 0.758 for female-faller, 0.751 for male-faller, 0.657 for female-nofaller, and 0.505 for male-nofaller. Faller was more unstable than nofaller and the male and female subgroups showed significant differences (*p* < 0.005).

The results of the calculation of the difference in stability between the joint points of the faller group and the no-faller group are shown in Table 3. The anterior superior iliac spine point in the hip joint showed a significant difference(*p* < 0.005).

### 3.4. Baseline Comparison Based on Machine Learning

The performance of the six machine learning classifiers is shown in Table 4 and Figure 5. The overall range of accuracy is 19.57~100%. In the faller group, the range of Precision is 0.1333~1, the range of recall is 0.2222~1, and the range of F1 score is 0.1667~1. In the no-faller group, the range of Precision is 0~1, the range of recall is 0~1, and the range of F1 score is 0~1. GBDT shows the best performance, with an rmse of 0 and an accuracy of 100%. In the faller group and the no-faller group, Precision, recall, F1 score are all 1.

## 4. Discussion

Data from 46 elderly people were analyzed using six machine learning classifiers, and the method predicted fall risk by identifying differences in stability at 42 joints in the human body. The GBDT classifier has good predictive performance, which is shown in Table 4. The model developed achieved acceptable accuracy in predicting the fall risk. To our best knowledge, this is the first study to incorporate multifractal algorithms, and machine learning algorithms to identify elderly people at high fall risk. The main findings were as follows: (1) In both the male and female cohorts, the faller group was less stable than the no-faller group; (2) the motion trajectories of the joints of the elderly were not regular periodic movements, as confirmed by the inverted bell-shaped fractal spectral curves, and the stability of the hip position was worse in elderly people with high fall risk; (3) the GBDT algorithm can be used as a useful tool to identify the fall risk in elderly. These findings are discussed in more detail below.

The multifractal algorithm is a new technology in signal analysis, image analysis, and other fields. It mainly studies the local laws and scale behavior of functions, describes the geometric distribution of function singularity, and is a function to measure the local regularity of signals from a global perspective. Due to the nonperiodicity and complexity of the motion trajectory, this study extends the multifractal algorithm to calculate the stability of the motion trajectory. By describing the characteristics of the generalized type spectrum, we use Dα Parameters to represent the stability of the subject’s motion trajectory.

The calculation results showed that all of the fallers were less stable than the no-fallers(*p* < 0.005). In the faller group, females had better stability than males; in the non-falls group, males had better stability. It has also been noted in previous studies that, unlike women, better motion ability in the male cohort did not help them to better prevent falls [32]. This may be caused by the decrease in judgment and decision-making ability of men during exercise with increasing age. Thus, in the falls group, men were more unstable; in the no-faller group, men were more stable and they adopted a more cautious approach.

By examining the cohort, we found that a significant decrease in the stability of the hip joint position occurred in people with a high risk of falls (*p* < 0.005). The hip joint plays a crucial role in maintaining the static stability of the body [33], with a greater contribution in restoring balance [34] (dynamic stability), and the location most likely to fracture after a fall occurs is also the hip joint; therefore, differences in stability regarding the hip position may explain the differences in fall risk.

Based on the interpretable analysis, further routine statistical analysis was conducted to determine whether there were significant differences between the groups in this study. As observed, interpretability is consistent with the results of statistical analysis. By studying the cohort, we found that those at high risk of falls experienced a significant decrease in the stability of the hip position (*p* < 0.005). The hip joint plays a crucial role in maintaining the body’s static stability [34], with a greater contribution in restoring balance [35] (dynamic stability), and it is also the location where fractures are most likely to occur after a fall, so differences in stability regarding the hip position may explain differences in fall risk.

In previous studies, researchers used many features of joints, such as the hip [33,34], knee [35], ankle [36], head [37], arm [38], etc. From the results, individual differences are common, which is one of the reasons why the most widely used fall risk screening tool is still the subjective scale. However, the group with high fall risk must have a decline in the stability of one or some joints, and this factor may be the reason for the increased fall risk.

After determining the feature dimensions, we used six machine learning classifiers (L1/2 sparse iteration, gradient boosting tree, SVM, random forest, RNN, DNN) to perform the prediction task. In this study, the accuracy of the GBDT classifier was 100%, in addition, the GBDT classifier had a Precision of 1, a recall of 1, and an F1-score of 1 in each class. Based on the 126-dimensional stability feature matrix we built, we believe that GBDT may be useful as a tool to identify fall risk in the elderly. In some previous studies, an accuracy of 67%~70% was obtained with the characteristics of step speed, stride length, and other factors [39]. One study obtained a precision of 80.7% using electronic health records of the elderly as a feature [40]. With the feature dimensions identified in this study, some improvement in prediction performance was achieved.

The clinical significance of the new method discussed in this work should be carefully considered. This method is based on the combination of interpretable machine learning algorithm and multifractal algorithm to determine the significant parameters that affect the fall risk of the elderly. In this study, we collected data through a fixed optical capture system, which is often used as the gold standard. With the development of wearable inertial sensing devices, there are many commercial sensing devices, which provide a lot of application scenarios for fall risk prediction. It is more convenient, and it can integrate video, audio, environment, and physiological signals to assess the gait of the elderly with different application scenarios and different types of diseases [41,42,43]. However, improving the convenience may also result in the loss of some joint point data. Therefore, in this study, we established an improvement of the Helen Hayes model [44], which expanded 24 points to 42 points, including all human motion joints as much as possible, and obtained the stability difference of all joint points, which also provides some data support for the development of wearable sensing devices. Developers can design sensing devices suitable for different positions of the human body according to the characteristics of the group.

The interpretability of multifractal algorithms or other similar tools is a key enabler to better understanding the decisions made by the black box model. However, the multifractal algorithm only gives the stability characteristics of different joint points, and the physiological factors that cause such stability differences need to be further judged by doctors. Therefore, the internal working principle of the model and the way of combining the characteristics to give the final decision are still implicit. In future work, we will integrate the use of EMG, inertial, video, and other sensors to determine the relationship between features and the possible direct and indirect impact of features on model output. This multimodal fusion method will enhance our understanding of the decision-making mechanism of machine learning algorithms, and these models will be used for more extensive fall risk screening and fall intervention.

Although the proposed algorithm has achieved significant performance in different literature and evidence, there are still certain limitations that need improvement, namely the limited amount of available data used. Furthermore, as we do not yet know what the possible physiological mechanisms behind the differences in the stability of fluctuation curves at different joints are, describing complex postural movements through an appropriate mathematical model (deterministic or stochastic) helps to address this issue. In future research, algorithm researchers and physicians can work together to solve this problem and thus find effective strategies for determining fall risk using posture dynamics. Future work will need to validate and verify the reliability of our results as there is not enough data available. The use of more data would greatly increase the credibility of the framework proposed in this study.

## 5. Conclusions

In this study, the motion stability of the human body was quantified in 42 positions and three dimensions by multiple fractal algorithms. The *D*(*α*) values showed significant differences (*p* < 0.005) in different groups for the determination of fall risk, and the motion stability of the nofaller group was better than that of the faller group in all cases. In addition, good prediction results were achieved by using the 42 × 3 feature matrix as the input dimension of the machine learning method, where the GBDT algorithm showed good performance. For future work, analyzing a larger faller queue with hyperparameter optimization would make it possible to have good performance in a wider range of applications.

The machine learning strategy makes full use of the motion data collected in the laboratory and is not limited to a single moment or segment of data. Due to the fact that we quantify the posture data of all joint positions in the human body, it is easier to help clinicians find the reasons for the increased risk of falls in the elderly. As the number of data increases, machine learning strategies are increasingly being incorporated into high-fall risk phenotypes to provide a personalized view of postural variability as part of the forward movement of risk.

## Figures and Tables

**Figure 1 healthcare-11-00047-f001:**
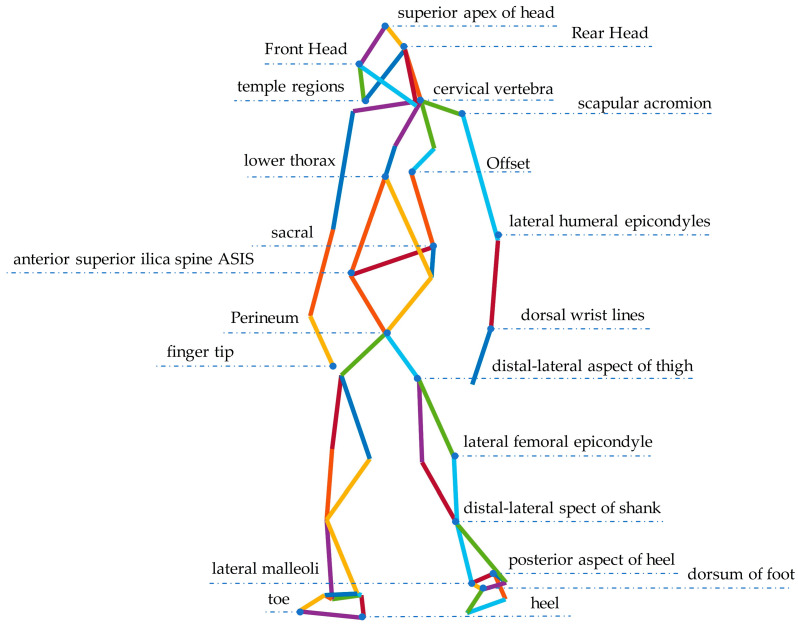
42-Markers model. (Some of the markers in this calibration model are symmetric and identified separately in Figure 1; the non-symmetric markers are identified entirely.).

**Figure 2 healthcare-11-00047-f002:**
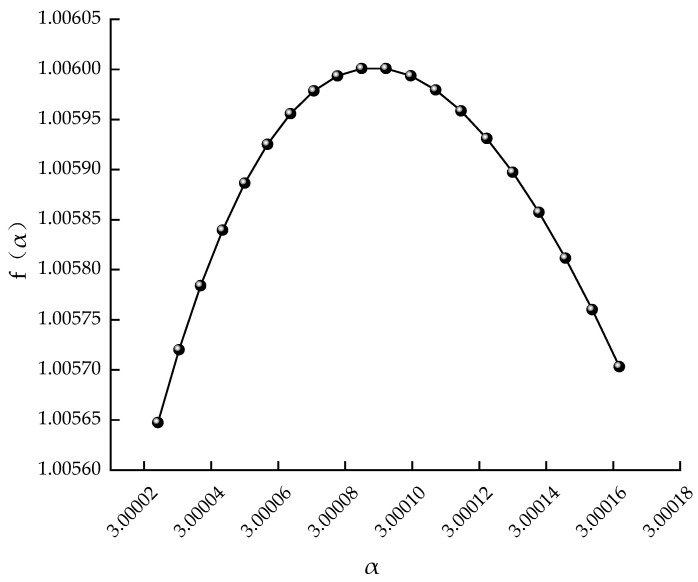
The generalized fractal spectrum of time series.

**Figure 3 healthcare-11-00047-f003:**
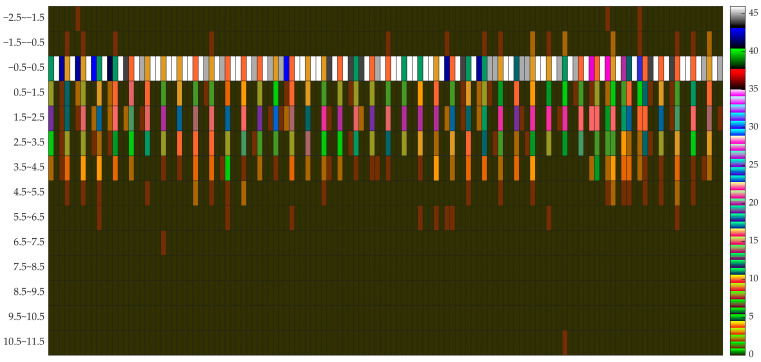
Stability calculation results (horizontal coordinates from left to right, a total of 126 cells, respectively, for the 1st to the 126th respectively, 42 markers in x-plane, y-plane, and z-plane stability results, the order is consistent with the serial number of Table 3).

**Figure 4 healthcare-11-00047-f004:**
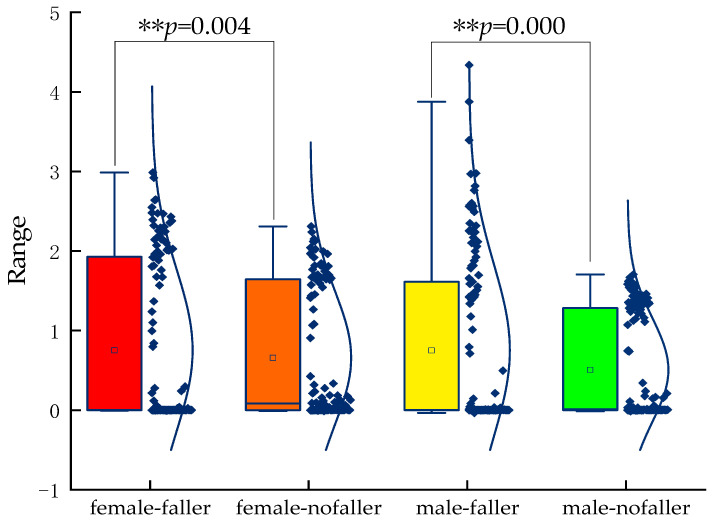
Statistical differences between groups. ** denotes significance (*p* < 0.005).

**Figure 5 healthcare-11-00047-f005:**
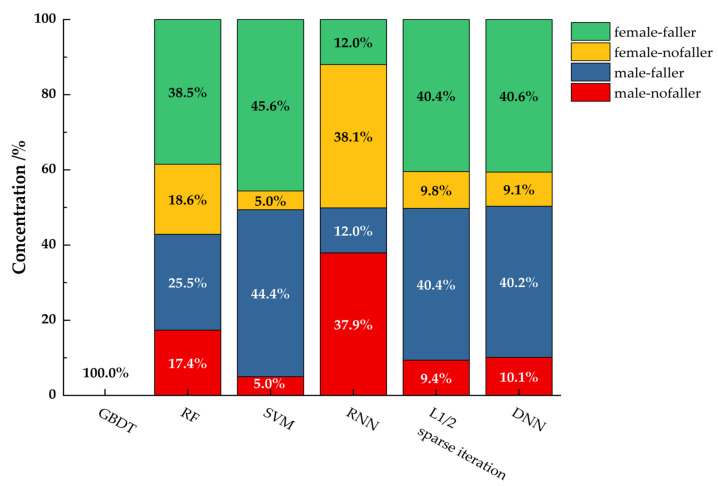
Prediction effects of machine learning algorithms in different groups (Note: All the RMSE of GBDT are 0).

**Table 1 healthcare-11-00047-t001:** Baseline characteristics.

Group		Age	BMI	Number of Falls		
				≥2	≥1, <2	=0
faller	Male	73	24.3	0	1	——
	Female	69 ± 5.10	23.2 ± 2.79	1	7	——
nofaller	Male	71.9 ± 7.18	25.7 ± 3.18	——	——	17
	Female	71.6 ± 7.44	25.6 ± 3.01	——	——	20

**Table 2 healthcare-11-00047-t002:** Parameter Setting.

Classifier	Parameter	Setting
L1/2 sparse iteration	Regularization parameter	0.1
	Kernel function	Polynomial kernel functions
	Maximum number of iterations	100
	Sparsity	3
	Kernel parameter	2
	Iteration step	0.1
SVM	Regularization parameter	0.01
	Kernel functions	Polynomial kernel functions
	Kernel parameter	1
GBDT	Loss function	Square Error
	Maximum depth	5
	Convergence tolerance	0.01
	Iteration step	0.1
	Number of decision trees	20
RF	Binning	32
	Maximum depth	5
	Number of decision trees	20
DNN	Number of hidden layer neurons	4
	activation function	RELU
	Regularization parameter	0.01
	Learning Rate	0.01
	Minimum number of batches	16
	Initial weighting	Xavier
	Number of iteration rounds	30
	Optimization function	Random gradient descent
RNN	Number of hidden layer neurons	4
	activation function	RELU
	Regularization parameter	0.01
	Learning Rate	0.01
	Minimum number of batches	16
	Initial weighting	Xavier
	Number of iteration rounds	30
	Optimization function	Random gradient descent

**Table 3 healthcare-11-00047-t003:** The significant differences between different markers.

Serial Number	Markers	Plane	*p*
1	R ^1^ dorsum of foot	x-plane	0.570845
2		y-plane	0.885085
3		z-plane	0.858193
4	L ^2^ dorsum of foot	x-plane	0.100503
5		y-plane	0.446369
6		z-plane	0.777331
7	R posterior aspect of the heel	x-plane	0.130772
8		y-plane	0.951891
9		z-plane	0.68468
10	L posterior aspect of the heel	x-plane	0.695843
11		y-plane	0.605083
12		z-plane	0.323039
13	R lateral malleoli	x-plane	0.031245
14		y-plane	0.598111
15		z-plane	0.493452
16	L lateral malleoli	x-plane	0.026079
17		y-plane	0.234059
18		z-plane	0.605729
19	R distal-lateral spect of the shank	x-plane	0.572168
20		y-plane	0.159774
21		z-plane	0.950345
22	L distal-lateral aspect of shank	x-plane	0.571991
23		y-plane	0.401798
24		z-plane	0.642101
25	R lateral femoral epicondyle	x-plane	0.27025
26		y-plane	0.977317
27		z-plane	0.62918
28	L lateral femoral epicondyle	x-plane	0.668132
29		y-plane	0.436801
30		z-plane	0.620824
31	R distal-lateral aspect of the thigh	x-plane	0.441128
32		y-plane	0.641446
33		z-plane	0.041118
34	L distal-lateral aspect of the thigh	x-plane	0.08325
35		y-plane	0.525894
36		z-plane	0.723975
37	R anterior superior ilica spine ASIS	x-plane	0.015412
38		y-plane	0.353301
39		z-plane	0.570597
40	L anterior superior ilica spine	x-plane	0.001171
41		y-plane	0.688984
42		z-plane	0.621601
43	sacral	x-plane	0.053304
44		y-plane	0.615821
45		z-plane	0.390852
46	R scapular acromion	x-plane	0.452359
47		y-plane	0.761281
48		z-plane	0.510924
49	L scapular acromion	x-plane	0.119001
50		y-plane	0.389242
51		z-plane	0.521805
52	R lateral humeral epicondyles	x-plane	0.200636
53		y-plane	0.522867
54		z-plane	0.040595
55	L lateral humeral epicondyles	x-plane	0.252793
56		y-plane	0.578391
57		z-plane	0.550779
58	R dorsal wrist lines	x-plane	0.113835
59		y-plane	0.460773
60		z-plane	0.927966
61	L dorsal wrist lines	x-plane	0.025517
62		y-plane	0.602451
63		z-plane	0.613074
64	The superior apex the of head	x-plane	0.698663
65		y-plane	0.005398
66		z-plane	0.652218
67	R temple regions	x-plane	0.083902
68		y-plane	0.744137
69		z-plane	0.936313
70	L temple regions	x-plane	0.11686
71		y-plane	0.142111
72		z-plane	0.639534
73	R medial malleoli	x-plane	0.234998
74		y-plane	0.253999
75		z-plane	0.54275
76	L medial malleoli	x-plane	0.515557
77		y-plane	0.787848
78		z-plane	0.710334
79	R medial femoral epicondyles	x-plane	0.863009
80		y-plane	0.459006
81		z-plane	0.233966
82	L medial femoral epicondyles	x-plane	0.05002
83		y-plane	0.614371
84		z-plane	0.61275
85	A ^3^ lower thorax	x-plane	0.403436
86		y-plane	0.39908
87		z-plane	0.607267
88	P ^4^ lower thorax	x-plane	0.680081
89		y-plane	0.622078
90		z-plane	0.612459
91	A cervical vertebra	x-plane	0.565881
92		y-plane	0.724782
93		z-plane	0.548561
94	P cervical vertebra	x-plane	0.389123
95		y-plane	0.295323
96		z-plane	0.04119
97	Perineum	x-plane	0.393137
98		y-plane	0.636805
99		z-plane	0.607256
100	R toe	x-plane	0.50184
101		y-plane	0.705765
102		z-plane	0.488071
103	L toe	x-plane	0.074054
104		y-plane	0.327039
105		z-plane	0.862776
106	R heel	x-plane	0.078012
107		y-plane	0.605572
108		z-plane	0.720744
109	L heel	x-plane	0.854931
110		y-plane	0.925216
111		z-plane	0.807634
112	R finger tip	x-plane	0.027352
113		y-plane	0.512236
114		z-plane	0.578072
115	L finger tip	x-plane	0.637861
116		y-plane	0.592343
117		z-plane	0.50689
118	Front Head	x-plane	0.435147
119		y-plane	0.497381
120		z-plane	0.627139
121	Rear Head	x-plane	0.088662
122		y-plane	0.12797
123		z-plane	0.626796
124	R Offset	x-plane	0.759189
125		y-plane	0.63617
126		z-plane	0.567955

^1^ “R” means right. ^2^ “L” means left. ^3^ “A” means anterior. ^4^ “P” means posterior.

**Table 4 healthcare-11-00047-t004:** Performance evaluation indexes of different classifiers.

Classifier	Accuracy	Precision		Recall		F_1_-Score	
		Faller	Nofaller	Faller	Nofaller	Faller	Nofaller
L1/2 sparse iteration	60.87%	0.2857	0.88	0.6667	0.5946	0.4	0.7097
SVM	97.83%	0.9	1	1	0.973	0.9474	0.9863
GBDT	100%	1	1	1	1	1	1
RF	93.48%	0.75	1	1	0.9189	0.8571	0.9577
DNN	56.52%	0.1333	0.7742	0.2222	0.6486	0.1667	0.7059
RNN	19.57%	0.1957	0	1	0	0.3273	0

## Data Availability

The data used in this study have been desensitized and do not contain identifying information about the subjects. The corresponding author may disclose upon reasonable request.

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
