# Peer review of "The Identification of Elderly People with High Fall Risk Using Machine Learning Algorithms"

_healthcare, 2022, doi:10.3390/healthcare11010047_

Round 1
Reviewer 1 Report
- The abstract should be rewritten. From my point of view, the object, method, and results should not be highlighted.
- How do you justify that GBDT is better than any other ML algorithms? The standard metrics for classification algorithms is accuracy, precision, recall as well as F1 score. These metrics should all be evaluated.
- The labels of the data are based on the falling history. How could you guarantee the correctness of the labels?
- More state of art research should be reviewed. And the contribution of this paper should be highlighted.
Reviewer 2 Report
The aim of this study was to analyze the relationship between postural stability ability and fall risk in an elderly population, and to construct a machine learning framework that can automatically identify fall risk in the elderly based on this study. The proposed machine learning strategy makes full use of the motion data collected in the laboratory and is not limited to a single moment or segment of data.
Although the work is technically sound and has comprehensive experimentation, there seems to be a missing gap on ground truthing for purposes of validating the proposed algorithm. Actually, tracking and classifying human activities by using IMUs are fully researched and there are popular commercial systems such as Xsens MVN (https://www.xsens.com/products/xsens-mvn-analyze/).
I value the authors efforts to present their significant research work. It seems to me that the subject matter is presented in a comprehensive manner. Overall, the paper quality largely meets the requirements of "healthcare", hence the manuscript could be accepted for publication after revision.
1. Literature review could be more focused. Maybe there lacks detailed explanations about the key contributions. The readers need more help to understand what is important, what is new, and how it relates to the state of art.
2. What are the limitations of the proposed identification method? The author did not mention it in the paper. What are the implications of the findings? More discussion should be provided in the manuscript. The factors that influence the accuracy of classification should be analyzed in more detail in the discussion section.
3. It would be nice to discuss the following papers in the paper:
- Celik, Y., Stuart, S., Woo, W. L., Sejdic, E., & Godfrey, A. Multi-modal gait: A wearable, algorithm and data fusion approach for clinical and free-living assessment. Information Fusion, 2022, 78, 57-70.
- S. Qiu, H. Zhao, N. Jiang, Z. Wang, L. Liu, Y. An, H. Zhao, X. Miao, R. Liu, G. Fortino. Multi-sensor information fusion based on machine learning for real applications in human activity recognition: State-of-the-art and research challenges, Information Fusion, 2022,80:241-265
- Talitckii, A., Kovalenko, E., Shcherbak, A., Anikina, A., Bril, E., Zimniakova, O.,Somov, A.. Comparative Study of Wearable Sensors, Video, and Handwriting to Detect Parkinson’s Disease. IEEE Transactions on Instrumentation and Measurement. 2022, 71, 2509910
4. Transitions from section to section should be smoother.
5. Proofread the paper and improve readability.
Reviewer 3 Report
The article is interesting. The recommendations have been added to the text of the attached manuscript.

Author Response
请参阅附件。

Reviewer 4 Report
The aim of this study was to develop a computational framework for the automatic identification of fall risk in older people using machine-learning algorithms. This fact can be a tool that helps to assess the healthcare in the elderly, helping to find the elderly with a higher risk of falls.
Why was gender not included in the initial basic information parameters?
The limit to know if one group belongs to the group of falls and the other to the group of no falls is not clear.
How was the sample calculated? Do you think it is representative of the population?
The limitations of the study have not been included.
Round 2
Reviewer 2 Report
The aim of this study was to analyze the relationship between postural stability ability and fall risk in an elderly population, and to construct a machine learning framework that can automatically identify fall risk in the elderly based on this study.
I value the authors efforts to answer all the previous concerns and feel that the paper overall improved. It seems to me that most of the previous concerns were well addressed in the revised manuscript. Overall, the revised paper quality largely meets the requirements of "Healthcare", hence the manuscript could be accepted for publication.
Reviewer 3 Report
The authors improved the article according with the recommnedations.